# Integration of Computational Pipeline to Streamline Efficacious Drug Nomination and Biomarker Discovery in Glioblastoma

**DOI:** 10.3390/cancers16091723

**Published:** 2024-04-28

**Authors:** Danielle Maeser, Robert F. Gruener, Robert Galvin, Adam Lee, Tomoyuki Koga, Florina-Nicoleta Grigore, Yuta Suzuki, Frank B. Furnari, Clark Chen, R. Stephanie Huang

**Affiliations:** 1Department of Bioinformatics and Computational Biology, University of Minnesota, Minneapolis, MN 55455, USA; 2Department of Experimental and Clinical Pharmacology, University of Minnesota, Minneapolis, MN 55455, USAleeam@umn.edu (A.L.); 3Department of Pediatrics, University of Minnesota, Minneapolis, MN 55455, USA; rgalvin@umn.edu; 4Department of Neurosurgery, University of Minnesota, Minneapolis, MN 55455, USAsuzuki.yuta@mayo.edu (Y.S.);; 5Department of Medicine, University of California San Diego, La Jolla, CA 92093, USA; ffurnari@health.ucsd.edu

**Keywords:** glioblastoma multiforme, drug discovery, biomarker discovery, drug response prediction, precision medicine

## Abstract

**Simple Summary:**

The major impact from this work is two-fold, first in providing a list of reproducible candidate drugs nominated against multiple glioblastoma (GBM) datasets, and second in providing drug–gene biomarkers of interest for GBM as well as reproducible computational pipelines for identifying drug–biomarker leads that could be extended to other cancer types beyond GBM. We believe the research community will be interested and can utilize our results here for the further development of biomarker-specific drug therapies in both GBM as well as other disease types.

**Abstract:**

Glioblastoma multiforme (GBM) is the deadliest, most heterogeneous, and most common brain cancer in adults. Not only is there an urgent need to identify efficacious therapeutics, but there is also a great need to pair these therapeutics with biomarkers that can help tailor treatment to the right patient populations. We built patient drug response models by integrating patient tumor transcriptome data with high-throughput cell line drug screening data as well as Bayesian networks to infer relationships between patient gene expression and drug response. Through these discovery pipelines, we identified agents of interest for GBM to be effective across five independent patient cohorts and in a mouse avatar model: among them are a number of MEK inhibitors (MEKis). We also predicted phosphoglycerate dehydrogenase enzyme (PHGDH) gene expression levels to be causally associated with MEKi efficacy, where knockdown of this gene increased tumor sensitivity to MEKi and overexpression led to MEKi resistance. Overall, our work demonstrated the power of integrating computational approaches. In doing so, we quickly nominated several drugs with varying known mechanisms of action that can efficaciously target GBM. By simultaneously identifying biomarkers with these drugs, we also provide tools to select the right patient populations for subsequent evaluation.

## 1. Introduction

Glioblastoma multiforme (GBM) is one of the deadliest diseases, with only approximately 2% of GBM patients surviving three years or more [1,2]. Following radiation therapy and surgery, standard of care for GBM can consist of chemotherapy such as temozolomide (TMZ) used for radiosensitization. Carmustine may also be used in certain situations such as in recurrent GBM [3]. Unfortunately, a large portion of GBM patients do not respond to these treatment strategies and develop resistance to them very quickly. There is an urgent need to develop new therapeutic options for GBM patients. Yet, the heterogeneous nature of this disease and the requirement of passing through the blood–brain barrier for most of systematic treatment have significantly limited the progress on development of efficacious drug treatment for GBM. Furthermore, traditional drug development takes on average about 12 years and 1 billion dollars to bring a new drug through regulatory approval. At this rate, hundreds and thousands of GBM patients will continue to suffer from lack of treatment. To this end, computational approaches to nominate drugs and identify biomarkers may offer a new path to significantly expedite the drug development process. 

In this study, we apply a computational pipeline, oncoPredict [4], to enable drug sensitivity prediction in patient tumors [5]. Previously, when applied to several hard-to-treat cancer settings (e.g., triple-negative breast cancer and castration-resistant prostate cancer), oncoPredict has successfully nominated and pre-clinically validated drugs currently undergoing animal and clinical evaluation [6,7]. We are now applying this computational tool to the GBM setting in order to nominate efficacious therapy and help combat this highly heterogeneous disease. Specifically, the computational pipeline was applied to eight independent GBM and non-high-grade glioma (non-HGG) patient datasets (spanning nearly 2000 patients total). In addition, we generated drug sensitivity predictions and performed experimental validation in a previously validated GBM mouse avatar model [8] to validate candidate drugs nominated across the GBM clinical cohorts. Furthermore, we employed a causal inference framework to identify biomarkers for the candidate drugs of interest with the goal to facilitate selection of the right patients to be treated with the right drug, which is critically important in heterogeneous disease like GBM [9]. Experimental validation of a selected biomarker was also carried out in the mouse avatar model. Our GBM avatar model, enriched with computational insights, has not only confirmed the efficacy of selected drug candidates but has also paved the way for more personalized treatment strategies in the relentless fight against GBM.

## 2. Materials and Methods

### 2.1. GBM Clinical Data Tested in Computational Modeling

In this study, we applied computational drug and biomarker discovery pipelines to nine publicly available primary low-grade glioma (LGG) and adult GBM patient datasets (Appendix A) to identify compounds of interest for specific patient populations defined by the presence of biomarkers. In total, our drug imputation model was applied to almost 2000 LGG and GBM samples, imputing almost 500 drug response scores for each sample. The LGG datasets consisted of bulk RNAseq from TCGA (*n* = 516) and the Chinese Glioma Genome Atlas (CGGA) (*n* = 282). TCGA datasets were downloaded using the TCGAbiolinks R package [10], and CGGA datasets were downloaded from the CGGA webpage [11]. The GBM datasets include microarray data (*n* = 332) and bulk RNAseq (*n* = 165) from TCGA, microarray data from CGGA (*n* = 102), microarray data from Rembrandt (*n* = 189, GSE108476) [12], and a personally combined dataset of published literature (referred to as CMDAT) also using affymatrix (*n* = 62). All datasets were previously normalized and log transformed to yield Gaussian expression distributions.

### 2.2. GBM Mouse Avatar Model Utilized for Experimental Validation of Drug Candidates and Inferred Drug–Biomarker Relationships

The GBM avatar model [8] was previously found to be a viable preclinical model for GBM. It was created by introducing different genetic driver mutations using CRISPR-Cas9 into human induced pluripotent stem cells. This was followed by differentiation of GBM-associated mutations containing neural progenitor cells (NPCs) and animal orthotopic engraftment to develop human adult GBM models. The NPC samples represent a pre-HGG state. Tumor cells were obtained and cultured to produce spheres, and this process was repeated twice to produce primary and secondary sets of tumors and spheres. Secondary tumors were obtained following engraftment of primary spheres, and secondary spheres were obtained from secondary tumors. This process is reflected in Appendix A. This resulted in 66 total samples including NPCs, engrafted tumor avatar samples, and spheres with technical replicas. Specifically, 6/66 samples were NPCs, and the remaining 60 samples consisted of 28 mesenchymal subtype samples and 32 proneural subtype samples. The samples representative of the mesenchymal subtype were characterized by PTEN and NF1 deletion, and the proneural subtype was characterized by TP53 deletion and PDGFRA mutation. The bulk RNAseq expression profiles for these NPCs, tumors, and spheres were obtained, and batch effect correction was performed where appropriate using remove unwanted variation (RUV) normalization [13].

### 2.3. Overview of Drug Discovery Pipeline

To identify drug leads or drugs predicted to elicit a greater response in GBM samples relative to non-high-grade glioma (non-HGG) samples, we applied a drug discovery pipeline. This pipeline involved comparing drug response scores or imputed AUC across different sequencing platforms (microarray and RNAseq) as well as patient and avatar data. Cancer cell line screening data have been used to train machine learning models, aiming to translate in vitro drug response to in vivo tumor response predictions and generate novel drug discovery hypotheses. To date, the Broad Institute’s Cancer Therapeutics Response Portal [14] (CTRP) is one of the largest publicly available drug screening efforts, providing drug screening for nearly 1000 cell lines and 500 compounds. The most updated dataset from CTRP is CTRP version 2 (CTRP2), representing a variety of cancers and molecular targets. CTRP’s cell line transcriptome data are provided through the Broad Institute’s Cancer Cell Line Encyclopedia database [15] (CCLE). For our purposes, the names across the cancer cell lines from CTRP2 and CCLE were harmonized to Cellosaurus accession numbers, indicated by the ‘CVCL’ prefix [16]. Drugs screened across <40% of all the cancer cell lines were also removed, helping to ensure robust predictions. This resulted in 887 cell lines and 493 drugs. Our R package oncoPredict’s function calcPhenotype() estimates a gene’s weight in determining a cell’s drug sensitivity through applying linear regression with a ridge penalty. This allowed a predictive drug sensitivity score, in the form of AUC to be obtained for each sample running through oncoPredict. Therefore, the imputed AUC was generated using a score system trained on high-throughput drug screenings in hundreds of cancer cell lines at a concentration range from 0 to 10 µM for all drugs (e.g., CTRPv2). These scores are in the general range of 10–20. To compare cross platform drug response data, we accounted for technical variation and platform effects by transcriptome integration using Rank-In [17] prior to running calcPhenotype().

The non-parametric Wilcoxon rank sum (WRS) tests were selected for the statistical comparison between GBM and non-HGG data. WRS tests were performed, as the assumption of normality did not hold for the independent samples *T*-test, in which case the WRS holds power advantages. To reduce the probability of making one or more false discoveries or type 1 errors, which are common in multiple hypothesis tests, the statistical Benjamini and Hochberg FDR controlling procedure was implemented to adjust *p*-values. Drugs were filtered for statistically significant *p*-values, which confirmed that differences in predicted drug response scores existed between GBM and non-HGG data, and ranked by their effect size. The Hodges–Lehmann estimate (HLE) was measured for the magnitude of significance or effect size, known as the location shift [18]. It established directionality to determine whether a given drug was recommended for GBM over non-HGG, offering a robust measure of effect size against outliers and distribution assumptions. Once drugs were ranked, the top percentage of significant drugs with the largest location shift were selected. The top 50th percentile of compounds with the largest HLE was selected as a conservative choice, as it was large enough to help prevent efficacious drugs from slipping through the pipeline while restrictive enough to help capture compounds with the greatest magnitude in response. Due to the high variability of drugs recommended for each GBM patient dataset, drugs recommended across the majority of clinical datasets were selected for comparison with those recommended across the avatar data. Drug imputation was also performed for the avatar dataset, which served as computational verification prior to experimental testing. Drugs selected in the top 50% of HLE across both the majority of clinical data as well as avatar data were brought forward as drug candidates for experimental validation.

### 2.4. Overview of Biomarker Discovery Pipeline 

Our biomarker discovery pipeline was employed to predict biomarkers for drug leads in an effort to uncover vulnerable patient populations. This pipeline utilized the SCC as a univariate approach to biomarker discovery with Bayesian-network learning (MMPC and hybrid-PC) [19] as a multivariate approach. Univariate approaches like the SCC are defined as those that evaluate the informativeness of each gene individually in isolation from the other genes according to a criterion. MMPC and hybrid-PC algorithms depict a more biologically accurate representation of gene interactions by testing for conditional independence. They implemented the Fisher Independence Test to measure the partial correlation coefficient between predicted drug response scores and the expression of a gene under scrutiny while conditioning on a set of the other predictor genes known as the ‘conditioning set’. The TCGA-GBM RNASeq dataset was selected as the primary GBM dataset used in the biomarker discovery pipeline because relative to microarray data, RNAseq has higher specificity. It can more accurately detect differential expression as well as rare or low expressed genes, and it outperforms microarray in determining transcriptomic characteristics of cancer [20,21]. The first step in this pipeline was to filter the TCGA-GBM RNASeq patient gene expression dataset for genes that were significantly (FDR *p*-value < 0.05) and moderately to strongly correlated (|SCC| ≥ 0.60) with the predicted drug response scores under scrutiny. The SCC was computed by setting ‘cc = TRUE’ in oncoPredict’s calcPhenotype() function, adjusted for Spearman correlation. 

After filtering by the SCC, we applied the MMPC algorithm to predict PCs. Filtering the gene expression data using SCC reduced the dimensionality of the MMPC algorithm’s input data. This achieved reproducibility of its outputs since Bayesian network learning algorithms like MMPC can suffer from variable order dependence, which is a problem with high-dimensional data like gene expression [22]. The partial correlation computed for a given gene is equal to the correlation between two sets of residuals when linear regression is applied: the first is between the drug under scrutiny and the conditioning set, and the second is between the gene under scrutiny and the conditioning set. Hence, this test regresses both the target and the variable under scrutiny on the conditioning set. Drugs which are predicted to be independent have a partial correlation of zero, larger values indicate greater dependency, and smaller values indicate limited dependency. MMPC outputs a test statistic, taking into account the directionality of the partial correlation, measuring the strength and the directionality of the predicted gene–drug associations identified. As the independence tests are performed, they progressively exclude irrelevant genes (genes that are independent from the predicted drug response scores). The end result of this algorithm is genes that have survived these elimination stages known as PCs. PCs will have significantly large test statistics, where the sign of the test statistic (whether it is positive or negative) indicates directionality. This sign allowed us to determine whether a gene under scrutiny was associated with drug sensitivity through up or down regulation. The MMPC algorithm does not distinguish parental and child nodes from the PCs identified, so we also applied the Hybrid PC algorithm to the filtered gene expression data to infer the structure of the Bayesian network between gene expression and predicted drug response in order to determine which PCs were parental nodes. By utilizing these univariate and multivariate approaches, we were able to predict correlation, causality, and which genes lead to drug sensitivity and how (whether through up or down regulation). 

### 2.5. Experimental Testing of Drug Candidates GBM Mouse Avatar Model

The avatar models used to computationally validate drug leads were also used for experimental evaluation. Drug leads (PD318088, trametinib, selumetinib) and standard of care agents (TMZ, carmustine) were tested in six avatar samples. Two samples were NPCs, two were primary spheres, and two were secondary spheres. Mesenchymal and proneural avatar samples were represented equally amongst these sample types. To measure efficacy of drug leads, CellTiter Glo assay was performed independently four times for each model and each drug with six technical replicates included in each biological test. The experimental concentration range employed was 0.015 nM–100 nM. The relative ATP was measured on day three post drug/control treatment. The concentrations used for testing were selected by referencing dose–response curves obtained from several published cell line drug screening datasets for GBM cell lines, reported in our Simplicity application. Doses selected are provided in Appendix A. For each drug tested, relative ATP was captured six times per sample across nine different doses. Drug response or measured AUC was obtained by generating dose–response curves Then measuring AUC using the trapezoidal rule to compute the area underneath the relative ATP curve through the R function trapz() from the package pracma. 

### 2.6. Experimental Testing of Inferred Drug–Biomarker Relationships in GBM Mouse Avatar Model

Through applying our biomarker discovery pipeline to TCGA-GBM RNAseq data, we hypothesized that PHGDH knockdown contributes to trametinib sensitivity and overexpression contributes to resistance. To experimentally test this relationship in GBM mouse avatar samples, secondary mesenchymal and proneural spheres with PHGDH knockdown and overexpression were treated with trametinib, and the surviving percentage was measured and compared to negative controls. Negative controls consisted of parental samples and empty vector transduced samples. To assess knockdown, control samples also included lentivirus plasmid vector pLKO.1-puro control vector.

## 3. Results

### 3.1. Identification of GBM Therapeutic Susceptibilities Utilizing Drug Response Prediction

In our drug discovery pipeline, we first applied oncoPredict to patient tumor transcriptome profiles to project patient likelihood of response to hundreds of medications. The patient tumor drug sensitivity projection was performed in five independent GBM patient datasets (a total of *n* = 850), a mouse GBM avatar model (*n* = 60), as well as two low-grade glioma (LGG) datasets from the Chinese Glioma Genome Atlas (CGGA-LGG, *n* = 282) and The Cancer Genome Atlas (TCGA-LGG, *n* = 516) and avatar neural progenitor cells (NPCs, *n* = 6). Details of datasets employed are listed in Appendix A. Our goal is to identify which drugs are repeatedly predicted to be more efficacious in GBM samples when compared to non-GBM samples across various datasets and modality. Hodges–Lehmann estimate (HLE) was performed with predicted drug sensitivity between GBM and non-GBM samples. Consensus across the majority (at least three of the five) of the GBM clinical datasets and validation in the mouse GBM avatar data relative to NPCs was set to select candidate drugs of interest. After comparing predicted drug sensitivity scores between each of the five GBM patient datasets and CGGA-LGG dataset, with these filtering criteria, we identified 22 drugs of interest (red * marked in Figure 1A, where HLE is also indicated in Figure 1B). The difference in these predicted drug response scores, indicated by the HLE, for these 22 drug candidates between GBM and non-HGG groups are shown in Figure 1B. When a different set of non-GBM control, TCGA-LGG, was used, we identified 62 drugs (red * marked in Appendix A, where HLE is also indicated in Appendix A).

Of the drug candidates predicted to be efficacious in GBM relative to non-HGG, six of these drugs were predicted to confer efficacy in GBM relative to both CGGA-LGG (Figure 1) and TCGA-LGG (Appendix A) non-HGG patient controls. These six drugs included mitogen-activated protein kinase inhibitors (MEKis) PD318088 and trametinib, as well as BRD.K71935468 (inducer of reactive oxygen species), Fumonisin.B1 (inhibitor of ceramide synthase), ML203 (activator of muscle pyruvate kinase), and RITA (inhibitor of p53-MDM2 interaction). Overall, the drugs identified across the LGG controls fell under a variety of classes including MEKis, EGFR inhibitors (EGFRis), and VEGFR inhibitors (VEGFRis). Other targets include STAT3, BRAF, CDKs, etc. (Appendix A). Many of these drug leads have already been identified as potentially efficacious therapeutics in preclinical studies, supporting the validity of our computational projection. A few of these candidates have been tested in patients, yet none of them have been tested in patient populations guided by biomarker screening. As seen in Figure 1 and Appendix A, MEKis were repeatedly identified as drug leads across independent patient datasets. For example, higher predicted sensitivity towards trametinib was observed across GBM samples relative to non-HGG (regardless of the control datasets: CGGA-LGG, TCGA-LGG, or NPCs) where the heatmaps display similar effect size and directionality measured by HLE with an FDR-corrected *p*-value < 0.001. Therefore, we went forward with experimentally testing trametinib and additional MEKis selumetinib and PD318088 in the GBM avatar models. It is worth noting that while our drug discovery pipeline focused on candidate drugs predicted across the majority of clinical datasets with computational validation in avatar data, several other drugs were predicted to show higher sensitivity in all GBM clinical datasets; however, they were not also predicted to confer higher sensitivity in the avatar GBM samples. The independent nature of these patient datasets made the discovery interesting as well, and these drugs’ information can be found in Appendix A. 

### 3.2. The Efficacy of MEKis Was Validated in GBM Avatar Model 

Fundamentally, we found a number of MEKis showing higher predicted sensitivity in GBM across a number of independent datasets. We selected trametinib, selumetinib, and PD318088 for experimental testing in our GBM avatar model along standard of care agents: temozolomide (TMZ) and carmustine. After exposing to increasing concentrations of TMZ or carmustine, both standard of care agents produced a lower area under the dose response curve (AUC) for GBM cells relative to NPCs (although this finding is only statistically significant for carmustine, *p* < 0.05, Appendix A), Lower and significantly different AUC values were also observed upon treatment of GBM cells with all three MEKis evaluated. In Figure 2, we plotted both imputed and experimentally measured AUC for MEKis across avatar samples. All three MEKis performed as predicted across these samples, where GBM samples were significantly (FDR *p*-value < 0.001) more sensitive to each MEKi tested, producing a lower AUC value relative to the NPC control samples. 

### 3.3. Application of Causal Inference to Identify Biomarkers Indicative of MEKi Response

We employed a computational pipeline intended for causal inference in large omic data analysis for biomarker discovery for our drug of interest. This enabled us to predict drug–gene relationships for the MEKi drug candidates identified. Our pipeline contains two steps: 1. Spearman correlation coefficient (SCC) analysis, a univariate approach with a goal to filter for the most informative genes and 2. Bayesian-network learning, a multivariate approach consisting of the min-max parents–child (MMPC) and hybrid-parent and child (hybrid-PC) algorithms to infer and visualize causal biomarkers. As shown in Figure 3A, through SCC analysis between TCGA-GBM RNA-Sequencing (RNASeq) data and predicted drug response for each MEKi, independently, we identified genes whose expression correlated with drug sensitivity (FDR *p* < 0.05 with moderately–strongly |r| >= 0.60). This substantially reduced the dimensionality of the transcriptomic data from tens of thousands to hundreds or less and enabled reliable application of the Bayesian algorithms. Specifically, the data’s dimensionality was reduced to 239, 72, and 54 informative genes for trametinib, PD318088, and selumetinib, respectively. Using each of these gene sets, the MMPC algorithm predicted parent and child (PC) nodes representative of genes that either directly influence or are directly impacted by drug response, where 8–12 PCs were identified for each drug. The MMPC algorithm also computed a test statistic, indicating the magnitude of the partial correlation for each PC, where a larger correlation indicates a stronger causal relationship. The hybrid-PC algorithm was also directly applied to the filtered TCGA-GBM RNASeq data to predict 1–2 parental nodes for each drug. For both trametinib and PD318088, phosphoglycerate dehydrogenase enzyme (PHGDH) was predicted to be a PC by the MMPC algorithm and a parental node by the hybrid-PC algorithm. In addition, SH2B adaptor protein was also predicted to be a PC for selumetinib and PD318088 as well as a parental node for selumetinib, directly influencing MEKi response (Appendix A). 

### 3.4. PHGDH Expression Levels Help Inform MEKi Response

Our computational pipeline identified PHGDH as a parental node for two MEKis with the largest test statistic from conditional independence testing (Appendix A). The significant *p*-value and the large test statistic indicates a significantly strong causal relationship between PHGDH expression levels and MEKi response relative to the other PC nodes. In addition, in the univariate analysis, we observed a significant and positive correlation between PHGDH gene expression and predicted or measured trametinib response across our six GBM datasets (five patient cohorts and avatar) (Appendix A). Taken together, we hypothesized that PHGDH knockdown increases cellular sensitivity to trametinib, and overexpression leads to trametinib resistance. To test this hypothesis, experimental testing was carried out by manipulating the PHGDH expression levels in a collection of GBM avatar samples (both proneural and mesenchymal subtype cells). We had successful knockdown and overexpression of PHGDH in all GBM avatar samples (Figure 3B). The GBM cells with/without PHGDH manipulation were then treated with increasing concentrations of trametinib, and the surviving percentage was measured and compared to controls. Two control samples with unmanipulated PHGDH expression were used, including a parental condition. To assess knockdown, control samples also included lentivirus plasmid vector pLKO.1-puro control vector. The average of three short hairpin RNA (shRNAs) constructs with PHGDH knockdown demonstrated significantly increased sensitivity to trametinib (*p*-value < 0.0001) relative to both control samples (Figure 3C). To assess overexpression, control samples also included plasmid pLV-EF1. The PHGDH overexpression led to significantly increased resistance to trametinib (*p*-value < 0.0001) relative to both control samples as well (Figure 3D).

## 4. Discussion

Utilizing a computational drug sensitivity prediction tool independently across five GBM patient datasets (approximately 1000 samples), a GBM mouse avatar model, and two control non-HGG patient datasets, we identified a collection of drugs that were predicted to have a higher sensitivity in GBM relative to non-HGG (the complete list is provided in Appendix A). While many of our nominated drugs (close to 20) have been/are being evaluated in clinical studies against GBM, nearly half (approximately 40) of our discoveries have shown efficacy in preclinical experiments reported previously. All of these justify the validity of our computational approach. Furthermore, given the heterogeneity among these independent clinical studies, the consistency in our discoveries further justify their importance and presents numerous opportunities for follow-up studies. These drugs achieve tumor growth inhibition through a variety of mechanisms of actions, namely VEGFRis, EGFRis, and MEKis, which have a rich history of study in the context of GBM [23,24,25,26,27,28,29,30,31,32,33]. Other targets identified that frequently show up in clinical studies include mammalian target of rapamycin (MTOR) and cyclin-dependent kinases (CDKs). In addition, drugs targeting STAT3, reactive oxygen species (ROS), PAR1, PAK4, apoptosis proteins, and ubiquitin-specific protease 14 (USP14) are also among our top predicted efficacious candidates in treating GBM. All of them present potential new opportunities. The one unique outcome of our approach is that we identified agents that are known to act through various mechanisms of actions yet are all effective in the setting of GBM simultaneously. 

To dive in deeper for the candidate drugs of interest, we found a number of VEGFRis through our pipeline, namely avastin, axitinib, and lenvatinib. VEGFRis have a long track record in GBM applications and have been identified as providing promising therapeutic opportunity if improvement of biomarkers aids in advancing the clinical efficacy of this approach [23]. Avastin is an example of such an inhibitor, which has been investigated in preclinical and clinical studies for its potential to delay GBM tumor growth. It is currently approved in adult patients whose cancer has progressed after prior treatment (recurrent or rGBM) [24]. Other VEGFRis include axitinib and lenvatinib, which have shown success in pre-clinical studies [25,26]. A prospective phase I and II study is currently in place to assess the effectiveness of lenvatinib in combination with pembrolizumab for GBM (NCT05973903).

EGFRis were also repeatedly identified as efficacious for treating GBM across our pipeline. In fact, EGFRis have had varying degrees of success against GBM. For example, afatinib, an EGFR inhibitor that has shown effect against GBM in preclinical studies [27,28]. However, when tested in GBM patients (NCT00727506) either alone or in combination with TMZ, only very limited efficacy was observed in unselected patients [29]. Importantly, follow-up study showed that afatinib is only effective in selected patients harboring EGFRvIII mutation [27]. Given the heterogeneity commonly recognized for a disease like GBM, this example highlights the need for biomarker screening as a means to direct patients to their appropriate treatment to help clinical trial design in drug development and eventually in the combat against GBM. 

In our study, multiple MEKis were repeatedly projected to show higher efficacy in GBM samples regardless which control datasets were used. We experimentally validated all three MEKis (trametinib, selumetinib, and PD318088) for their preferential activity in mouse avatar samples when compared to NPC. It was worth noting that in the same experimental validation model, we also tested two standard of care agents (TMZ and carmustine), which show very subtle differences between GBM and NPC samples, suggesting potential resistance towards standard of care treatment. In this case, our candidate drugs, MEKis, showed more pronounced differences between the two groups, supporting their more selective efficacy in the GBM cells. Surveying the literature, we found that preclinical evidence supports the efficacy of trametinib and selumetinib in treating glioma. For example, trametinib has been reported to exert a strong antiproliferative effect on multiple established GBM cell lines, and its inhibitory effect on cell growth was observed even after standard of care treatment [30]. Selumetinib was reported to stabilize disease progression in glioma patients during a phase II clinical trial [31]. PD318088, on the other hand, is a more novel MEKi, which has not been clinically tested against GBM [32].

Causal inferences have been employed to study disease risks, yet constraint-based algorithms are yet to be widely used in the identification of drug biomarkers for GBM. In this study, we integrated multiple approaches for biomarker discovery for our drug of interest. We first apply a commonly used univariate analysis between gene expression and drug sensitivity to narrow down the list of informative genes, then two Bayesian causal inference tools utilizing constraint-based algorithms were employed to identify causal genes. Through this pipeline, we nominated PHGDH as our top MEKi biomarker and were able to experimentally validate its role in MEKi sensitivity. Specifically, we found PHGDH expression levels as having a significant and positive correlation and causal relationship with trametinib response. Interestingly, PHGDH has previously been proposed as a therapeutic target for melanoma in overcoming resistance to MEKis PD0325901 and trametinib [33,34], where suppression of PHGDH led to sensitivity in MEKi-resistant melanoma cells. These findings mirror the directionality predicted by our biomarker discovery pipeline, supporting that the same biomarker for MEKi may be utilized in other cancer settings. Indeed, we envision the same biomarker discovery pipeline can be applied to any other drugs and/or phenotypes in the future. In addition, our biomarker discovery pipeline identified several other genes which may directly influence MEKi response as well. For example, SH2B was predicted to be a PC for both selumetinib and PD318088 as well as a parental node for selumetinib, directly influencing MEKi response. It is highly expressed in GBM, promoting progression through activating STAT3 signaling [35]. It is known to play a critical role in promoting GBM progression and has previously been proposed as a new therapeutic target. Therefore, our discovery also warranted studying of this gene as a potential biomarker for MEKis. 

## 5. Conclusions

Overall, the findings from this study support the approach to integrate computational and experimental models as well as multiple independent cross platform and cross species datasets, with the goal to speed up drug discovery and development for GBM. Our research supports ongoing efforts to improve selectivity of treatments applied to patients with GBM through applying discovery pipelines to GBM expression data, pairing drug candidates with biomarkers indicative of drug response. 

## Figures and Tables

**Figure 1 cancers-16-01723-f001:**
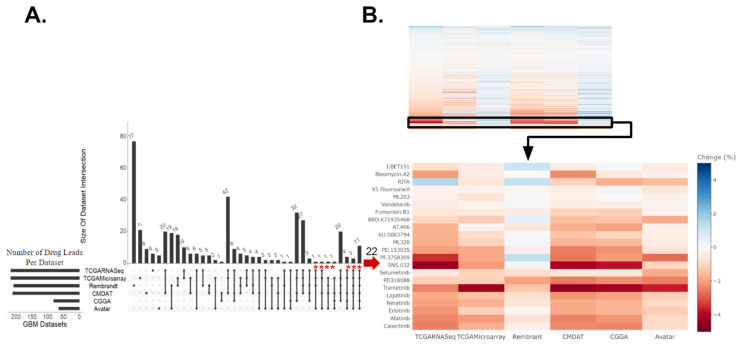
Drug candidates identified for GBM relative to non-HGG (CGGA-LGG and NPC). (**A**) Upset plot displaying intersections of drugs predicted to be efficacious for GBM relative to non-high-grade glioma (non-HGG) across six GBM datasets. These drugs had a Hodges–Lehmann estimate (HLE) within the top 50% of drugs with an FDR-corrected *p*-value of ≤0.05. The asterisk indicates 22 drugs that were predicted as efficacious in more than half of the clinical datasets against CGGA-LGG and validated in the avatar dataset, which included MEK inhibitors selumetinib, PD318088, and trametinib. (**B**) The top panel heatmap displays the standardized HLE for all drugs identified to have a significantly different drug response across GBM and non-HGG samples. The bottom panel heatmap displays the HLE specifically for the 22 drug leads. Drugs predicted to be more efficacious for GBM are darker, in red. Those more efficacious for non-HGG are lighter, in blue.

**Figure 2 cancers-16-01723-f002:**
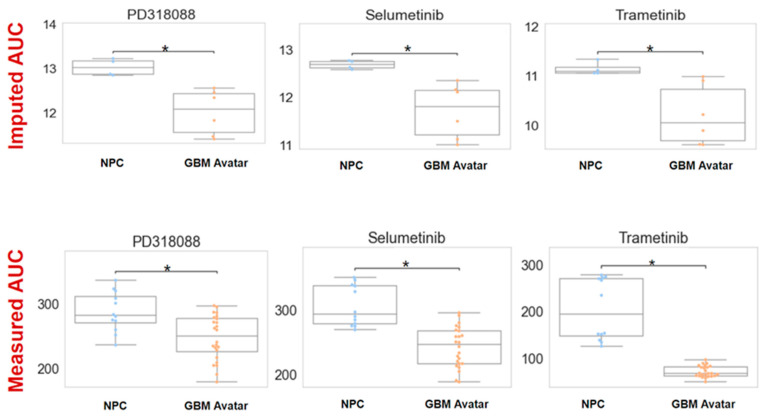
Imputed vs. measured drug response of MEK inhibitors (MEKis) across glioblastoma (GBM) avatar and control samples. Top row: Predicted drug sensitivity scores for three MEKis (PD318088, selumetinib, trametinib) obtained from oncoPredict. AUC (area under the dose–response curve) was predicted for neural progenitor cells (NPCs) and GBM samples for three MEKis. The asterisk indicates statistical significance (FDR *p*-value < 0.05) Bottom row: Measured drug response by exposing GBM avatar or NPC cells to each MEKi separately with increasing concentrations (across a range of 0.015 nM–100 nM for the drug). AUC was calculated using area under the dose–response curve after CellTiter experiments. For both imputed and measured scenarios, AUC across GBM and NPC samples were compared using a Wilcoxon sum rank test. The asterisk indicates statistical significance between NPC and GBM samples. For imputed drug response, the *p*-value was 0.006, 0.01, and 0.02 for PD318088, selumetinib, and trametinib, respectively. For measured drug response, *p* < 0.001. To ensure the accuracy and the reliability of the CellTiter experiments, four biological replicates with six technical replicates within each experiment were measured to minimize the variability and to ensure the accuracy.

**Figure 3 cancers-16-01723-f003:**
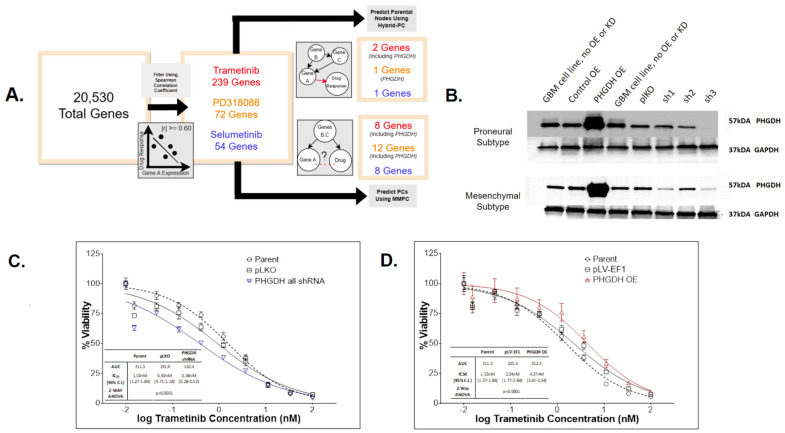
PHDGH was identified and validated to be a biomarker that affects MEKi treatment effect in GBM. (**A**) Biomarker discovery pipelines and findings for trametinib (red), PD318088 (gold), and selumetinib (blue). The pipeline consisted of filtering these data using the Spearman correlation coefficient (SCC) to select genes whose significant and absolute SCC was equal to or exceeded 0.60 (FDR *p* < 0.05) then applying two Bayesian algorithms (MMPC and Hybrid-PC) to infer correlation and causality between gene expression and drug response using these filtered data. From this pipeline, PHGDH expression was predicted to directly influence both trametinib and PD318088 responses. (**B**) Western blot of PHGDH and house-keeping control GAPDH in two GBM avatar subtype samples (the proneural and mesenchymal subtype, each carrying unique genomic modifications). The experiment confirms knockdown and overexpression of PHGDH were successful in our GBM avatar (across mesenchymal and proneural GBM subtypes). (**C**,**D**) CellTiter Glo experiments after trametinib exposure for 72 h in GBM avatar cells (of both proneural and mesenchymal subtypes) following PHGDH knockdown (**C**) and PHGDH overexpression (**D**). Each experimental condition was evaluated in triplicate, and the plots (**C**,**D**) represent the average across all experiments. For knockdown experiments, 3 sets of shRNA were employed, and their average effects were plotted as shown in the blue curve in panel C. A 2-way ANOVA test was employed to test the statistical differences among conditions. There are statistical differences in the cell survival after trametinib treatment between PHGDH knockdown and both controls (*p* < 0.001 for both parental and pLKO). There are also statistical differences in the cell survival after trametinib treatment between PHGDH overexpression and both controls (*p* < 0.001 for both parental and pLV-EF1). A significant increase in drug resistance occurred following overexpression as well as a significant increase in drug efficacy following knockdown.

## Data Availability

The original Western Blots of Figure 3B could be found via https://osf.io/ar9zg/ (accessed on 18 April 2024).

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
