# Peer review of "Integration of Computational Pipeline to Streamline Efficacious Drug Nomination and Biomarker Discovery in Glioblastoma"

_cancers, 2024, doi:10.3390/cancers16091723_

Round 1
Reviewer 1 Report
Comments and Suggestions for Authors
In this manuscript Maeser and colleagues propose an integration of computational pipeline to simplify drug nomination and biomarker discovery in Glioblastoma. Although the study is very interesting, there are some concerns which should be addressed from my point of view.
1. As one of the main messages of this article is "the power of the computational integration approach", authors should include at the beginning of the manuscript a schematic representation that clearly shows the applied computational pipeline.
2. Figure2. The authors mention that each sample was tested 6 times, which is only relevant for the imputed AUC, please provide details of the measured AUC data, there are big differences in the analyzed samples, including between NPC and GBM avatar, and these differences are not explained in the figure legend or in materials and methods section.
3. Figure 3. B.The authors should provide a quantitative measurement for the Western Blot data to see variability between the three experiments mentioned in the figure legend.
4. Figure 3. C.D. Authors should add error bars to the points from the surviving fraction curves because they have enough replicates to justify adding them.
Author Response
Dear Reviewers,
Thank you for your thorough review and valuable comments regarding our manuscript. We are grateful for the positive remarks about the potential value of our study. We greatly appreciate the opportunity to clarify and improve our study based on your insights. In the file attached, we have provided a point by point response to each of the specific points you’ve raised to further clarify our methods and findings.
Sincerely,
Stephanie Huang and Co-authors

Reviewer 2 Report
Comments and Suggestions for Authors
In this study, entitled ‘Integration of Computational Pipeline to Streamline Efficacious Drug Nomination and Biomarker Discovery in Glioblastoma’, the authors applied a computational pipeline to eight independent GBM and non-high-grade glioma (non-HGG) patient datasets. The drug sensitivity predictions were validated in a GBM avatar model. The study is worth adding to the literature and requires only minor corrections.
In the introduction, the standard treatment is explained as follows: "After radiotherapy and surgery, the standard treatment for GBM may consist of chemotherapy (carmustine) with temozolomide (TMZ) or transcranial magnetic stimulation Since TMZ is also known as a radiosensitising agent, this description is not entirely correct and needs to be revised.
The Introduction section needs a concluding sentence summarising the GBM avatar data.
Standard treatments were also tested in avatar models and showed no effect, as shown by the AUC. This may also be the case in patients who are resistant to these drugs. This should be mentioned in the discussion, which increases the clinical utility of this study.
It would be good to add a histological image for the avatar model that tracks GBM development in the brain or other data to confirm this. Perhaps also on the spheres?
Author Response

(The authors gave the same response as above.)
